# Severity of Omicron Subvariants and Vaccine Impact in Catalonia, Spain

**DOI:** 10.3390/vaccines12050466

**Published:** 2024-04-27

**Authors:** Víctor López de Rioja, Luca Basile, Aida Perramon-Malavez, Érica Martínez-Solanas, Daniel López, Sergio Medina Maestro, Ermengol Coma, Francesc Fina, Clara Prats, Jacobo Mendioroz Peña, Enric Alvarez-Lacalle

**Affiliations:** 1Department of Physics, Universitat Politècnica de Catalunya, Castelldefels, 08860 Barcelona, Spain; aida.perramon@upc.edu (A.P.-M.); clara.prats@upc.edu (C.P.); enric.alvarez@upc.edu (E.A.-L.); 2Public Health Agency of Catalonia, Department of Health, 08005 Barcelona, Spain; luca.basile_ext@gencat.cat (L.B.); sergiomedina@gencat.cat (S.M.M.); jmendioroz@gencat.cat (J.M.P.); 3Health Quality and Assessment Agency of Catalonia (AQuAS), 08007 Barcelona, Spain; ericamartinez@gencat.cat; 4Primary Care Services Information System (SISAP), Institut Català de la Salut, 08007 Barcelona, Spain; ecomaredon@gencat.cat (E.C.);; 5University of Vic—Central University of Catalonia (UVic-UCC), 08500 Vic, Spain

**Keywords:** SARS-CoV-2 variants, Omicron, COVID-19, epidemiology, COVID-19 vaccines, vaccine effectiveness, vaccination strategy

## Abstract

In the current COVID-19 landscape dominated by Omicron subvariants, understanding the timing and efficacy of vaccination against emergent lineages is crucial for planning future vaccination campaigns, yet detailed studies stratified by subvariant, vaccination timing, and age groups are scarce. This retrospective study analyzed COVID-19 cases from December 2021 to January 2023 in Catalonia, Spain, focusing on vulnerable populations affected by variants BA.1, BA.2, BA.5, and BQ.1 and including two national booster campaigns. Our database includes detailed information such as dates of diagnosis, hospitalization and death, last vaccination, and cause of death, among others. We evaluated the impact of vaccination on disease severity by age, variant, and vaccination status, finding that recent vaccination significantly mitigated severity across all Omicron subvariants, although efficacy waned six months post-vaccination, except for BQ.1, which showed more stable levels. Unvaccinated individuals had higher hospitalization and mortality rates. Our results highlight the importance of periodic vaccination to reduce severe outcomes, which are influenced by variant and vaccination timing. Although the seasonality of COVID-19 is uncertain, our analysis suggests the potential benefit of annual vaccination in populations >60 years old, probably in early fall, if COVID-19 eventually exhibits a major peak similar to other respiratory viruses.

## 1. Introduction

The SARS-CoV-2 virus has undergone numerous mutations, allowing the disease to continually evolve and adapt. Among various lineages, the Omicron variant, characterized by high transmissibility, emerged in Europe in late 2021. Several Omicron lineages —BA.1, BA.2, BA.5, and BQ.1—dominated in Europe throughout 2022 [1,2]. More recently, in 2023, new subvariants such as XBB.1.5, BA.2.86, EG.5.1, and JN.1 have emerged, reflecting the continuous adaptation of the virus in an ongoing global challenge [3,4,5]. 

On the other hand, at the end of 2020, following numerous experimental studies, several types of vaccines against coronavirus disease 2019 (COVID-19) became available and established themselves as an important and effective preventive measure [6,7,8]. The continuous evolution of the virus underscores the importance of adapting vaccination strategies to keep pace with viral changes and highlights the critical role of booster doses in enhancing immunity against emerging variants [9,10]. Real-life observational studies have demonstrated the protective effect of these vaccines in different global populations. In fact, although the high efficacy of COVID-19 vaccination in preventing hospitalizations and deaths due to COVID-19 was reported in the Alpha and Delta era (usually >80% for individuals fully vaccinated) [11,12,13], booster doses became necessary in the Omicron era to achieve similar levels of protection against severe disease [14,15,16]. Fortunately, despite its ability to partially evade vaccine-induced immunity, Omicron has been associated with reduced disease severity and lower hospital mortality compared to earlier variants [17,18]. In addition, efficacy in preventing SARS-CoV-2 infection was significantly lower and decreased over time, and evaluation of vaccine effectiveness beyond 6 months has been suggested as critical for updating vaccine policy [19]. Therefore, since the Omicron era, most studies have focused on the importance of timely vaccination regarding the risk of hospitalization and/or death, and thus its importance in terms of the public health impact of this infection [14,17,18].

In Spain, the COVID-19 vaccination campaign began on 27 December 2020, targeting the population from older to younger age groups. Catalonia, an autonomous community in northeastern Spain with a population of 7.7 million people, has observed trends similar to other European countries [20]. Throughout 2022, four waves of different variants have caused a significant increase in cases, hospitalizations, and deaths. In addition, during the study period, Spain, including Catalonia, implemented booster campaigns that included third and fourth doses. This provides a unique opportunity to study the impact of the vaccines on different variants and within diverse age groups.

The aim of this study is to evaluate the impact that booster doses of COVID-19 vaccine have had in reducing or avoiding hospitalizations and deaths due to different Omicron lineages in confirmed cases among the most vulnerable people. In addition, we try to establish a correlation between the emergence or dominance period of these variants and their severity in different age groups over 60 with varying vaccination status.

## 2. Materials and Methods

The Public Health Agency of Catalonia (ASPCAT) serves as the local health authority responsible for monitoring and responding to the pandemic. In Catalonia, the health system is public, universal, and free, ensuring equal access to health care for all residents. It updates public data on a weekly basis through the Infection Surveillance Information System in Catalonia (SIVIC) [20]. This system provides key metrics, including daily COVID-19 case numbers, current hospital bed occupancy, intensive care unit (ICU) admissions, and the distribution of virus variants. Daily case numbers are derived from an individual database of COVID-19 positive diagnostics, which are reported by both public and private health centers, including primary care and hospitals, to ASPCAT. This report follows the official Protocol for the Epidemiological Surveillance of COVID-19 (PESC) [21]. The data set includes detailed information such as dates of diagnosis, hospitalization, and death, the date of the last vaccination (whether an individual was fully vaccinated or had received a booster dose), and the cause of death. Due to laboratory saturation and the challenges in monitoring all cases during the Omicron peak in January 2022, the PESC criteria for diagnosis and notification were revised on 28 March 2022. The revised criteria prioritize cases in people over 60 years of age and other categories of vulnerable patients [22].

### 2.1. Review of the Epidemiological Data and Estimation of Variant Prevalence

We used the SIVIC public database [20] to track the prevalence of different SARS-CoV-2 variants in Catalonia from December 2021 to January 2023. During this period, Delta, BA.1, BA.2, BA.5, BQ.1, and XBB.1 variants predominated at different times, although other variants were also circulating residually. In Catalonia, COVID-19 variant surveillance is conducted through a mix of random and targeted sampling methods [23]. Detailed descriptions of the sampling proportions, methods, and criteria for targeted sampling are provided in Appendix A.

Figure 1 provides a comprehensive overview of variant prevalence over the course of the COVID-19 pandemic and the epidemiological situation in Catalonia during the study period, as presented in three distinct subplots. The top subplot shows the weekly variant counts, as sequenced by SIVIC, and described in Appendix A Appendix A. The middle subplot combines the weekly observed variant proportions from SIVIC with our mathematical model [24] to estimate the daily percentages of each variant. The bottom subplot uses these percentages calculated from the daily case data to display the estimated weekly case counts per variant and, in addition, to calculate the effective reproduction number, R_t_, employing the methods established in a previous study [25].

The output of the model allows us to identify days when a variant exceeds certain prevalence thresholds, allowing a thorough comparison of the number of cases, hospitalizations, and deaths attributable to each variant. Detailed information can be found in Appendix A Appendix A.

### 2.2. Epidemiological Trends: Hospitalizations and Deaths

The ASPCAT database complements the SIVIC database by providing individual patient follow-up, either by telephone or through the study of clinical records, and expands the information on hospitalization and mortality. Between 5 December 2021, and 26 January 2023, the ASPCAT database recorded 1,817,428 COVID-19 cases. Of these, 415,629 cases, classified as vulnerable, underwent thorough follow-up via epidemiological surveys. This process, which continued until either recovery or death, involved periodic phone calls to assess the status of the patient. For deceased patients, the database specifically categorizes the cause of death as due to COVID-19, with COVID-19, or unrelated to COVID-19, thereby providing valuable insights into the mortality trends and patterns associated with the virus. The age distribution of these closely monitored cases was as follows: 227,686 (<60 years), 66,104 (60–69 years old), 68,241 (70–79 years old), and 53,598 (>80 years old). Appendix A provides a time-based overview of age and vaccination status (Appendix A) and metrics on cases, hospitalizations, and deaths by age group (Appendix A). All numbers used in this study are available in Appendix A Appendix A for age groups 60–69, 70–79, and >80 years, respectively.

We excluded individuals under the age of 60 from our analysis for three main reasons. First, and most importantly, as noted above, since the protocol revision of 28 March 2022, active surveillance with epidemiological surveys has focused primarily on individuals over 60 years of age, a group that is easily identified as vulnerable. So, testing protocols in this group should not have significantly changed during the study period. Second, although younger individuals account for a significant number of cases, their impact on hospitalizations (14.3%) and deaths (2.8%) in the ASPCAT database is small. Nevertheless, it is important to recognize that these percentages are actually higher than what would typically be expected, even including the first wave of COVID-19 [26]. This discrepancy arises because the database, in line with the PESC criteria update, preferentially records data on more vulnerable groups, skewing the perception towards higher hospitalization and death rates among those under 60 years of age. These metrics, derived from 227,686 cases, indicate an accentuated representation of risk within this younger cohort compared to the general younger population. Third, the age group over 60 years, for which vaccination is fundamentally recommended, is of particular interest for assessing the impact of vaccination. In addition, the free and public nature of the Catalan health care system significantly reduces inequalities in access to health care among the study population. This minimizes the potential confounding effects of variations in access to health care on our results. As a result, our data set robustly reflects an older population that has been consistently, actively, and closely followed.

In addition, as we have the dates of diagnosis, hospitalization, and death, we excluded some inputs when comparing different metrics: when there were more than 14 days between diagnosis and hospitalization or more than 21 days between diagnosis and death. We based these exclusions on the assumption that longer intervals suggest a reduced direct association with SARS-CoV-2 infection, as confirmed in different studies in the literature, e.g., [27,28]. Table 1 summarizes the distribution of COVID-19 cases, hospitalizations, and deaths by sex, age group, variant, and vaccination status for those over 60 years of age, after the application of exclusion criteria.

### 2.3. Definition of Emergence and Dominance Periods

To assess the impact of the variants, we define two distinct time periods for each variant: (i) emergence, which begins when the prevalence of a variant exceeds 10% and continues until it surpasses 90%, and (ii) dominance, characterized by a variant maintaining a prevalence above 90% and then falling below 90% as another variant starts emerging.

### 2.4. Severity Metrics by Vaccination Status and Reduction in Severity Calculation

To study the impact of the vaccine on severe outcomes in confirmed test-positive cases, we stratified the data by age group and the time elapsed since the last vaccination dose, focusing on hospitalizations vs. cases, deaths vs. cases, and in-hospital deaths vs. hospitalizations. Monthly cohorts were aggregated to enhance statistical robustness, increasing sample sizes within each category. For an in-depth explanation, Appendix A provides a detailed example using the cohort aged over 80 as a case study, detailing procedures and presenting individual statistical outcomes.

To examine these percentages, the data are categorized into four different contingency tables, each representing cases, hospitalizations, deaths, and in-hospital deaths. Pearson’s $\chi^{2}$ tests are then performed on each category to assess their statistical independence. Due to the retrospective nature of this study, there is considerable variation in the data between metrics, age cohort, variants, and vaccination status. For this reason, we have considered statistical analysis methods to indicate those points that could not provide valuable information for clinical interpretation.

First, we calculated a rate ratio (RR), defined as the fraction of the proportion of severe outcomes (hospitalization, death, or in-hospital death) to the COVID-19 reference inputs (cases, cases, and hospitalizations, respectively) for the vaccinated group compared with that for the unvaccinated group [29]. To quantify the uncertainty, 95% confidence intervals were derived using the Clopper–Pearson exact method appropriate for the binomial distribution model. Fisher’s exact test was then used for each outcome to assess the association between vaccination status and disease severity, yielding *p*-values indicating non-random associations such as the confidence interval. Finally, we calculate what we call the reduction in severity as 1−RR [29], analogous to some methods of estimating vaccine effectiveness using confirmed COVID-19 cases [30,31,32]. However, it is important to note that these methods are not equivalent to an analysis of vaccine effectiveness that includes a control group, since the denominators are different; therefore, our results will generally show lower values. We also use standard error propagation to obtain 1−RR errors.

## 3. Results

### 3.1. Severity during Emergence and Dominance Periods of Variants

We analyzed data on individuals aged over 60 years, focusing on the daily number of cases, hospitalizations, and deaths attributed as due to COVID-19 during the emergence of Omicron subvariants BA.1, BA.2, BA.5, and BQ.1. To correctly assign severity measures within the appropriate variant period, hospitalization and death are assigned to the day of COVID-19 diagnosis rather than the day of hospitalization or death. A visual overview of the evolution of severity and the introduction of the different subvariants can be found in Appendix A. The analysis shows the correlation between the increase in variants BA.1 and BA.5 and the increase in daily cases, hospitalizations, and deaths.

Table 2 presents the results averaged over the two previously defined time periods: emergence and dominance, with the emergence period color-coded in green and red to denote whether the values are higher or lower, respectively, compared to the dominance period for the same variant. An extended version of this table is provided in Appendix A Appendix A. Furthermore, to examine the daily fluctuations of the same metrics presented in Table 2 or Appendix A, Appendix A provides an analysis of the daily changes in these severity metrics, confirming that all results consistently point in the same direction.

### 3.2. Cases, Hospitalizations, and Deaths Stratified by Vaccination Status

The preceding analyses provide a broad overview of the severity associated with different Omicron variants in Catalonia. Notably, the timelines for these analyses correspond to two booster dose campaigns conducted in Spain in late 2021 and fall 2022. Building on this, we now consider the impact of vaccination on severe cases. Again, in terms of mortality, we will only consider deaths due to COVID-19.

Performing Pearson’s $\chi^{2}$ tests on cases, hospitalizations, deaths, and in-hospital deaths yields substantially high $\chi^{2}$ values and, consequently, low *p*-values. This suggests that the timing of vaccination and the circulating Omicron subvariants are correlated.

Figure 2 shows all the results for the percentage of hospitalizations versus cases. It also shows deaths vs. cases and in-hospital deaths vs. hospitalizations for individuals aged over 80, who constitute the highest number of exitus events. Data points are differentiated by symbols representing vaccination status: 1–3 months (○), 4–6 months (□), 7–9 months (∆), more than 10 months (∇), and never vaccinated (◊). The 95% confidence intervals shown clearly delineate the associations between outcomes across variants and age groups, similar to the *p*-value from Fisher’s exact test. Detailed quantitative data can be found in Appendix A Appendix A. Moreover, empty symbols indicate that data for a particular outcome, such as cases or hospitalizations, are derived from a small number of counts/events. This assessment is based on the minimum sample size that would be required for a prospective study to include all cases, hospitalizations, and deaths in each age group. Given the retrospective nature of our data collection, we know the number of each outcome in advance, which influences the calculated probabilities; therefore, these points are not discarded but indicated. As a result, although Figure 2 shows all outcomes, caution should be exercised in interpreting these white-dashed points due to the high potential for statistical error (see Appendix A Appendix A for details).

Appendix A provides a more detailed explanation of Figure 2 and adds figures for the results for deaths and in-hospital deaths for age groups 60–69 and 70–79. In addition, extended analyses support the trends observed in Figure 2. These supplementary figures consolidate and confirm the patterns across all age groups by combining data from all variants studied.

### 3.3. Reduction in Severity

From previous findings presented in Figure 2, we further examined the impact on vaccination as the reduction in severity, 1−RR, against various subvariants and among specific age cohorts.

Figure 3 shows the trend for the reduction in severity in preventing hospitalizations and deaths due to COVID-19, segmented by three age cohorts, and evaluated according to the time elapsed since the last vaccine dose. Omicron subvariants and vaccination timing follow the same color and symbol scheme as in Figure 2. Each point represents a measure of the effectiveness of the vaccine in infected cases as a function of the rate ratio, 1−RR, referenced to cases or hospitalized individuals with a positive SARS-CoV-2 test within a specific post-vaccination interval. Solid symbols represent statistically significant results compared to the unvaccinated group, as indicated by the corresponding *p*-values, while open symbols indicate results that are not statistically significant. For better visualization, results with error bars greater than 100% are not plotted. Error bars represent the 95% confidence intervals for each measure. Results for deaths are shown only for the 80+ cohort due to insufficient statistical significance for other age groups.

In Appendix A extends the analysis of Figure 3 to include more results for the youngest groups, adding the evaluated significance that follows the pattern of the previous figure. Similarly, Appendix A confirms the observed trend of vaccine efficacy over time in these age groups by aggregating all data across variants; these are consistent with the main findings.

## 4. Discussion

This retrospective study conducts a thorough analysis of the epidemiological dynamics of multiple SARS-CoV-2 Omicron subvariants in Catalonia. A major strength lies in our methodology, which distinguishes between the emergence and dominance periods of the variants, revealing that the emergence period typically exhibits higher hospitalization and mortality rates. Another strength of our work is the use of a highly reliable population-wide database with individual records of COVID-19 events, vaccinations, and well-documented causes of hospitalization and death. This database covers all of Catalonia over a period that includes the evolution of the Omicron subvariants BA.1, BA.2, BA.5, and BQ.1 and spans two national vaccination campaigns. Using this data set, we were able to rigorously assess the severity of different variants in confirmed positive cases of SARS-CoV-2 according to the time since the last vaccination across different age groups.

Our findings highlight the increased severity observed during the emergence periods of BA.2, BA.5, and BQ.1 compared to their periods of dominance. The data indicate that the average daily number of cases and deaths for variants BA.5 and BQ.1 are consistently higher during their emergence periods than during their dominance periods. The same pattern holds for hospitalizations due to variant BA.5. These increases vary depending on age and the metric studied. For variant BA.2, both hospitalizations and deaths are also higher during the emergence period, although the difference is minimal. However, this trend does not hold for cases in individuals over the age of 70. This variation may be due to the evolution of SARS-CoV-2, the intrinsic characteristics of BA.2, or both. BA.2 emerged as daily cases declined from the peak of BA.1. Moreover, the slow emergence of BA.5 may have led to early cases, hospitalizations and deaths being misattributed to the later dominance of BA.2 by our data analysis methods. This discrepancy could also be attributed to the potential protective effects of prior BA.1 infection [33], which could reduce the occurrence of more severe cases. For BA.1, unlike other variants, the emergence period saw higher hospitalizations and deaths than the dominance period only in the 60–69 age group. The first Omicron variant, BA.1, which emerged in December 2021, reported the highest case numbers of the entire pandemic and had a transmissibility significantly higher than Delta [24,29]. This led to an incredibly rapid emergence period for this variant, approximately only three weeks, and during the subsequent dominance month, numbers hovered around the peak numbers. This explains why the dominance period outnumbers the emergence period. Finally, if we examine the average differences in hospitalization rates between the BA.* variants, we find trends similar to those in [34]. For individuals over 60 years of age, BA.2 and BA.5 were associated with a 12% lower and a 21% higher risk of hospitalization, respectively, compared with BA.1. In contrast, [34] reported a 15% lower and 18% higher risk of hospitalization. Despite comparing different time periods, in which daily numbers of cases detected may change due to different tracking of asymptomatic or mildly symptomatic cases and may be potentially misleading, universal access to the health system and the homogeneity of testing protocols in patients older than 60 should minimize any such possible bias. Qualitatively, our percentages are expected to be higher because our data include only at-risk individuals confirmed by COVID-19 testing, resulting in a naturally higher hospitalization-to-case ratio.

The results show a consistent pattern across all age groups and variants for the severity of confirmed positive cases: recent vaccination significantly reduced the likelihood of the disease worsening. This trend is consistent across the three age groups studied, although the magnitude of the effect varies by variant and age group. This same observation has been supported by several studies, e.g., [35,36,37], but such stratification by subvariant and/or age group was not found in these studies. The hospitalization results present a robust set, despite small samples and large confidence intervals in some of their vaccination status results, supporting the majority of analyses with statistically significant findings. In particular, for hospitalizations related to the BA.1 and BA.2 variants, the differences in outcomes between 1–3 months and 4–6 months post-vaccination are only marginally distinct. However, in cases where these differences are statistically significant (BA.1 in the 60–69 and 70–79 age groups and BA.2 in the 60–69 age group), the more recently vaccinated individuals have better outcomes. This pattern remains consistent for BA.1, BA.2, and BA.5 variants when comparing results between those vaccinated 7–9 months ago and those vaccinated more than 10 months ago. Importantly, the results for vaccinated people are always better than those for unvaccinated people for all four variants. Finally, there are fewer statistically significant results for mortality, with BA.1, BA.2, and BA.5 showing clear differences only for the over-80 age group. Although only BA.1 shows a clear difference between those vaccinated at different times, all the results show how vaccination quantitatively protects older individuals against fatal outcomes compared to those unvaccinated.

Our results on preventing hospitalizations for the BA.1, BA.2, and BA.5 variants and across all age groups show that the more recent the vaccination, the higher the effectiveness in infected cases. However, there was no evidence of reduced effectivity against hospitalization for BA.5 compared to BA.2 or BA.1; this is similar to the findings in [38]. We found that vaccine impact against severe disease decreased significantly by 3 to 4 months, which was consistent with [39,40]. Although the trend is the same in all studies, we cannot directly compare the magnitudes because our severity reduction study was conducted with confirmed COVID-19 cases, whereas the previous studies were conducted with a test-negative case-control group. In the case of BQ.1, unlike the previous variants, very few real-world studies have focused exclusively on this variant and the efficacy of vaccines. We know that the BQ.1 variant has enhanced immune evasion capabilities compared to previous omicron variants [41] but, nevertheless, booster doses have a positive effect against hospitalizations caused by this variant [42]. Studies in England show that this protection, which reaches a maximum within the first month, declines slightly after 10 or more weeks, but remains at a plateau after 6 months for monovalent vaccines [42,43]. Interestingly, in our study we cannot distinguish what happens during the first 12 weeks, but we found that the efficacy of the vaccine for the BQ.1 variant seems to remain relatively constant regardless of the time elapsed since the last dose, in line with the plateau mentioned in [43].

Considering all the factors discussed and the results obtained, and assuming that COVID-19 could develop a stable, annual seasonal pattern similar to that of other respiratory diseases, the vaccination schedule for vulnerable groups could be optimized. However, the exact seasonal trend of COVID-19 has not been conclusively determined; it is uncertain whether we will see a single annual peak, multiple peaks within a year, or some other pattern. Current data from the 2021–2022 and 2022–2023 seasons in Europe, corresponding to periods of dominance of different Omicron subvariants, show several peaks, two of which are more significant: one in late fall or winter and another in late spring or summer, depending on the conditions of each region. The first peak, corresponding to colder temperatures, had a significantly greater effect [44,45,46]. Given this uncertainty about the temporal symmetry and relative magnitude of future peaks, it is not feasible to propose a precise vaccination strategy now. Nevertheless, if COVID-19 eventually follows a winter-dominated seasonality similar to influenza, our results suggest that the optimal timing for vaccination campaigns in Catalonia would be early fall (with the highest peak of infections around December–January [47]), following the same vaccination pattern as for influenza, although different regions could require different timetables according to the stationary seasonality. However, if COVID-19 adopts a biannual peak pattern, the vaccination strategy should be discussed based on the relative severity of each peak and focused on the most vulnerable individuals. This approach will certainly require new studies with updated data.

It is important to acknowledge the limitations of our retrospective study. First, due to the surveillance nature of the database, we were only able to work with COVID-19 positive cases; the results for hospitalizations and deaths should therefore be understood exclusively for this population subgroup and should not be directly compared with, e.g., vaccine effectiveness results from other studies. Similarly, because this is a retrospective study and we did not calculate the minimum sample size beforehand, all results are shown with their *p*-values for the comparable results. For example, we only discussed the deaths in the over-80 age group because they were more numerous. In any case, all figures and numbers can be found in Appendix A. In addition, changes in the surveillance protocol (March 2022) may have led to unequal detection between the BA.1/BA.2 and BA.5/BQ.1 variants. However, these limitations are unlikely to have a significant impact on our findings because our study focused on individuals aged 60 years and older, who were subject to more rigorous and consistent follow-up and diagnostic protocols. Another important limitation is the lack of detailed information on comorbidities and previous infections in the available data set, due to the nature of the surveillance design. This absence of data on comorbidities, which are known to affect the severity of COVID-19 outcomes, and on previous infections, which may influence immunity levels, represents a gap that prevents a thorough analysis of the vaccine impact on vulnerable populations based on real-world data. In addition, the lack of available data on reinfections and the resulting adjustments in vaccination schedules hinders a comprehensive assessment of their impact on vaccine effectiveness and protection levels. Recognition of these limitations underscores the importance of interpreting our findings within the context of the available data.

## 5. Conclusions

Our study sheds light on the complex epidemiology of the SARS-CoV-2 Omicron subvariants in Catalonia. Through temporal segmentation, we observed that the emergence period is generally more severe, characterized by a higher number of hospitalizations and deaths. A key observation underscored the increased efficacy of vaccinations received within the last six months in reducing the number of severe outcomes. Moreover, there is a notable difference between the results in the 1–3 month and 4–6 month intervals for the 60–69 and 70–79 age groups. Furthermore, while the BA.1, BA.2, and BA.5 variants suggest that the effect of the vaccine is higher in more recently vaccinated individuals, the constant effect pattern observed for the BQ.1 variant, regardless of the time elapsed since the last dose, is noteworthy and requires further investigation.

Finally, it is important to highlight the practical implications and interpretative value of our study for public health strategies and future vaccination policies. Our results suggest that periodic vaccination remains an important tool to reduce hospitalizations and deaths, especially in individuals over 60 years of age. Consistent with the window of effective protection observed in the first three months after vaccination, we propose that, in the event that COVID-19 ultimately shows a seasonality pattern similar to that of other respiratory viruses, with a major peak in the coldest months, the booster vaccine would be administered once a year in early fall. In this way, we would optimize protective measures before the expected seasonal wave, providing greater protection against the most severe cases. If this is not the final scenario, and COVID-19 shows a different seasonality pattern, new studies will be needed to address its dynamics and its implications.

## Figures and Tables

**Figure 1 vaccines-12-00466-f001:**
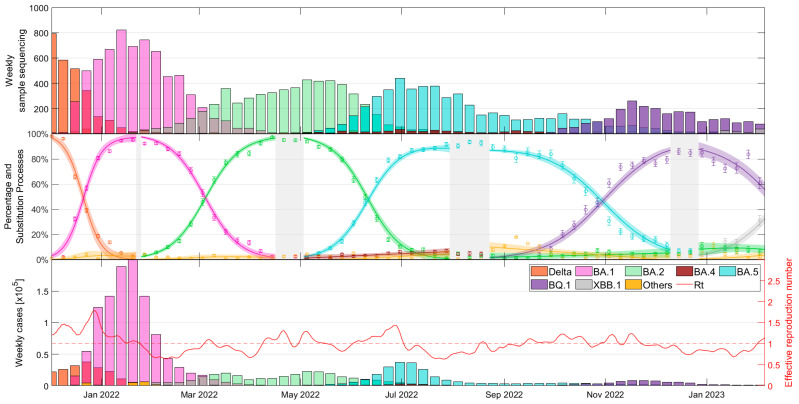
Dynamics of COVID-19 in Catalonia. (**Top**) Weekly counts of analyzed variants, (**middle**) observed and modelled variant proportions, and (**bottom**) estimated weekly cases by variant and calculated effective reproduction numbers (R_t_) based on daily case counts.

**Figure 2 vaccines-12-00466-f002:**
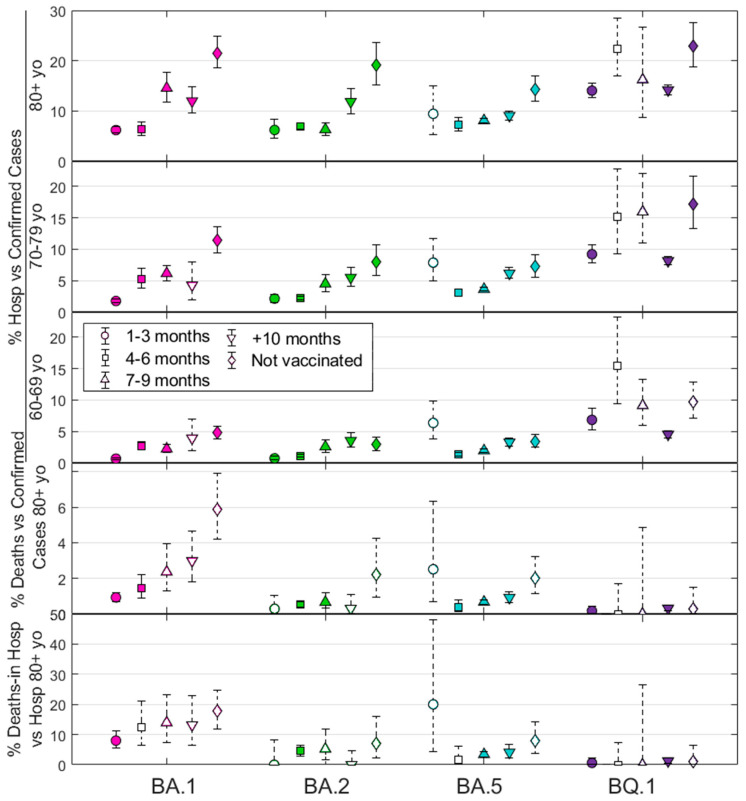
Five plots show the percentage metrics of hospitalizations relative to total cases for three age cohorts, deaths relative to total cases, and in-hospital deaths relative to new hospitalizations for individuals aged 80 years and older. The four different Omicron subvariants (BA.1, BA.2, BA.5, and BQ.1) are shown in different colors. Symbols represent different post-vaccination periods and non-vaccinated individuals (refer to the legend). Error bars show the 95% confidence interval. Empty symbols indicate a small number of events, suggesting the need for caution in making premature conclusions from these data points.

**Figure 3 vaccines-12-00466-f003:**
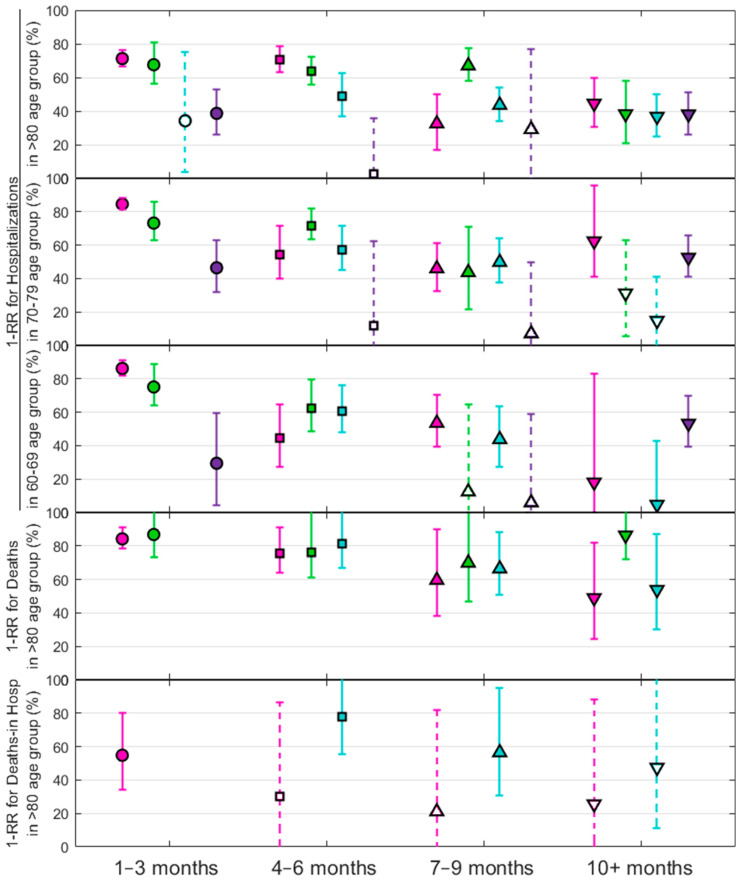
Reduction in severity for hospitalizations against four Omicron subvariants—BA.1 (pink), BA.2 (green), BA.5 (blue), and BQ.1(purple)—across three age groups over different post-vaccination intervals (top three plots). Reduction in severity for deaths and in-hospital deaths (fourth and fifth plots) is shown only for the cohort over 80 years of age across different post-vaccination intervals. Solid symbols represent statistically significant values, while open symbols indicate non-significant results compared to the unvaccinated group.

**Table 1 vaccines-12-00466-t001:** Distribution of COVID-19 cases, hospitalizations, and deaths classified as due to COVID-19 segregated by sex, age group, SARS-CoV-2 variant, and vaccination status.

	Cases		Hospitalizations		Deaths Due to COVID-19	
	*n*	%	*n*	%	*n*	%
By sex						
Male	80,138	42.7	5079	55.7	434	52.6
Female	107,584	57.3	4045	44.3	391	47.4
By age group						
60–69	66,002	35.2	1415	15.5	61	7.4
70–79	68,135	36.3	2800	30.7	151	18.3
80+	53,585	28.5	4909	53.8	613	74.3
By variant						
BA.1	39,980	21.3	1511	16.6	269	32.6
BA.2	47,829	25.5	1589	17.4	158	19.2
BA.5	75,636	40.3	3635	39.8	355	43.0
BQ.1	24,277	12.9	2389	26.2	43	5.2
By vaccination status						
<90 days	39,461	21.0	1351	14.8	131	15.9
90–179 days	49,643	26.4	1608	17.6	154	18.7
180–269 days	58,184	31.0	2853	31.3	283	34.3
>270 days	30,869	16.4	2419	26.5	136	16.5
Not vaccinated	9565	5.1	893	9.8	121	14.7

**Table 2 vaccines-12-00466-t002:** Average daily cases, new hospitalizations, and deaths for two different time periods per variant: (i) emergence and (ii) dominance.

	Average Cases/Day			Average Hospitalizations/Day			Average Deaths/Day (Due to COVID-19)		
	60–69	70–79	>80	60–69	70–79	>80	60–69	70–79	>80
BA.1 emergence	192.1	117.4	70.8	4.5	4.3	5.9	0.15	0.33	1.37
BA.1 dominance	287.3	282.0	198.4	4.1	8.2	16.0	0.14	0.77	2.97
BA.2 emergence	203.4	171.2	107.5	2.7	5.3	9.5	0.12	0.26	0.84
BA.2 dominance	201.8	217.6	143.5	2.6	5.2	9.2	0.00	0.20	0.77
BA.5 emergence	277.4	321.2	276.6	5.1	12.0	22.4	0.12	0.28	2.00
BA.5 dominance	46.1	53.7	56.4	1.7	3.1	5.9	0.04	0.09	0.53
BQ.1 emergence	77.4	92.6	79.9	3.3	6.7	9.9	0.03	0.08	0.28
BQ.1 dominance	31.9	37.7	45.5	3.2	5.9	9.6	0.00	0.00	0.00

## Data Availability

The individualized data set analyzed during the current study is available in anonymized format in the Public Health Agency of Catalonia (ASPCAT) and aggregated data are published in the Infection Surveillance Information System in Catalonia (SIVIC). The original classified data used in this manuscript are included in the supplementary material tables. The MATLAB codes used for the analyses in this study can be accessed at https://github.com/BIOCOM-SC/cloud-of-codes (accessed on 20 February 2024).

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
