# Peer review of "Severity of Omicron Subvariants and Vaccine Impact in Catalonia, Spain"

_vaccines, 2024, doi:10.3390/vaccines12050466_

Round 1
Reviewer 1 Report
Comments and Suggestions for Authors
The study “Severity of Omicron Subvariants and Vaccine Effectiveness in Catalonia, Spain” analyzed COVID-19 cases from December 2021 to January 2023, focusing on Omicron subvariants BA.1, BA.2, BA.5, and BQ.1. Results showed that recent vaccinations significantly reduced disease severity, with effectiveness waning after six months post-vaccination, except for BQ.1. Timely vaccination was emphasized to mitigate severe outcomes influenced by variant and vaccination timing.
Concerns:
The exclusion of certain age groups and subvariants due to small sample sizes, should be addressed comprehensively in the discussion to acknowledge potential biases and limitations in the findings.
The study could benefit from a more in-depth discussion on the implications of the results for public health strategies and vaccination campaigns, including recommendations for optimizing vaccination schedules.
The study should provide a more detailed analysis of the potential confounding factors that could influence the observed vaccine effectiveness, such as comorbidities, previous infections, or variations in healthcare access among different age groups.
Author Response
The authors are grateful for the comments of the two reviewers and have done their best to add, modify, and clarify the text of the original manuscript to make it more rigorous and scientifically relevant. The reviewers will find new text and modified paragraphs in red. In this document, the reviewers will find the responses to their comments one by one, the new manuscript and Suppl. Mat. Texts. The lines in the main manuscript where we have made changes directly related to the comments of the two reviewers are marked in the responses. Other parts of the manuscript that were indirectly changed by the comments are also marked in red.
Please find the detailed responses and the corresponding revisions and corrections highlighted in the re-submitted files.

Reviewer 2 Report
Comments and Suggestions for Authors
1. This dataset only includes data on cases of SARS-CoV-2 and not the whole population of older adults. Firstly, the authors need to define what they mean by 'vaccine effectiveness'. Usually effectiveness against hospitalisation uses the proportion of cases of hospitalisation amongst vaccinated/ proportion of cases of hospitalisation amongst unvaccinated. I think here they are perhaps using proportion of cases of hospitalisation amongst vaccinated covid cases/proportion of cases of hospitalisation amongst unvaccinated covid cases. Using the cases as the denominator seems likely to provide an under estimate of effectiveness as the vaccine would reduce both the denominator and the numerator in the vaccinated group.
2. The SIVIC captures 'positive diagnostics, which are reported by both public and private health centers' so it sounds like it will only include cases where the individual has actively gone to a healthcare provider for a test. There will be many more cases that go unreported and the proportion and type of cases that go unreported will have varied over time - for example there will have been times during the pandemic where everyone went for external testing and others where people used home tests or did not bother to test. Comparing different variants/time periods according to the number of cases per day seems potentially misleading.
3. The paragraph 'To validate the observed differences in disease severity by vaccination status, we calculated the statistical power and used Fisher's exact test in each of our results. This statistical tool aims to identify non-random associations between two categorical variables' is very unclear. Statistical power is something to be calculated in advance of a study and relates to the probability of being to detect a particular effect size. Perhaps the wrong word has been used here.
4. Similarly 'a minimum sample size, n_min, was established for each variant, age group, and vaccination status to define a threshold that must be exceeded for the 95% level of statistical significance' is very unclear. A minimum sample size would normally be linked to a particular effect size and power level - so this is perhaps what is missing - however this would be constant for all groups.
5. The asterisks and lines on figure 2 are very distracting and seem unnecessary given the confidence intervals - I'd recommend removing. The confidence intervals encourage readers to focus on the magnitude of differences and precision rather than the arbitrary significance thresholds. Hosp vs cases is a confusing label - is this 'proportion of cases that were hospitalised'? Again, this is only out of cases and not the population in general and the denominator will change according to both vaccination status and time since vaccination - conclusions need to reflect that - this study can not make conclusions relating to vaccine efficacy or timing of vaccine in my opinion.
Author Response

(The authors gave the same response as above.)

Round 2
Reviewer 2 Report
Comments and Suggestions for Authors
The manuscript has been greatly improved in my opinion. I have no further comments.